# Comparative Short- and Long-Term Effectiveness and Safety of Pramipexole and Aripiprazole Augmentation in Treatment-Resistant Unipolar Depression: An Observational Study

**DOI:** 10.3390/biomedicines12092064

**Published:** 2024-09-10

**Authors:** Antonio Tundo, Sophia Betrò, Rocco de Filippis, Roberto Felici, Chiara Lucangeli, Marica Iommi

**Affiliations:** 1Clinical Section, Institute of Psychopathology, 00196 Rome, Italy; sophia.bet@gmail.com (S.B.); roccodefilippis@istitutodipsicopatologia.it (R.d.F.); robertofelici79@gmail.com (R.F.); lucangeli.chiara29@gmail.com (C.L.); 2Center of Epidemiology Biostatistics and Medical Information Technology, Department of Biomedical Sciences and Public Health, Università Politecnica delle Marche, 60121 Ancona, Italy; m.iommi@univpm.it

**Keywords:** depressive disorders, outpatient psychiatry, pharmacotherapy, dopamine agonists, depression treatment, treatment outcome

## Abstract

Background: This study compares the short- and long-term effectiveness and safety of pramipexole augmentation (PA) and aripiprazole augmentation (AA) for unipolar treatment-resistant depression (TRD). Methods: Patients were recruited in a private out-patients clinic specializing in mood disorders. At intake and at each visit, depressive and (hypo)manic symptoms, clinical status, and level of functioning were evaluated with appropriate scales. The trend of outcomes was analyzed using mixed-effect linear regression models. Results: The study includes 81 patients with unipolar TRD treated with PA and 51 with AA. After 12 and 24 weeks of treatment with PA, the predicted response (64.1% and 76.2%) and remission rates (49.7% and 72.7%) were significantly higher than the predicted response (32.2% and 38.0%) and remission rates (18.9% and 28.1%) for AA. The improvement in psychosocial functioning was significantly greater and faster in PA than in AA. PA showed significant superiority over AA as a maintenance strategy (time spent ill and psychosocial functioning) up to 12 months. No difference in safety was found at each time point. Conclusions: PA could be an alternative option for the short- and long-term treatment of unipolar TRD, more effective than AA and similar in safety. These preliminary results need confirmation from randomized clinical trials.

## 1. Introduction

Major depressive disorder (MDD) is a common [1] and highly disabling [2] psychiatric disorder leading to substantial direct and indirect costs for the society [3]. Although several classes of antidepressants (ADs) of proven efficacy are available for treatment of major depressive episodes (MDEs), from 15% to 30% of patients do not respond to multiple interventions [4,5].

MDE not responding to two or more different ADs are referred to as treatment-resistant depression (TRD) [5], a condition associated with suicidal risk, poor prognosis, work impairment, social and family life impairment, and physical health decline [6].

Among the different pharmacological strategies proposed for the treatment of TRD, AD augmentation with a second-generation antipsychotic (SGA), mostly aripiprazole, currently has the best evidence of efficacy [7,8], is approved by the FDA, and is recommended by international guidelines [9,10,11]. Regulatory agencies, in consideration of the available evidence [12], also recently approved Esketamine nasal spray, an N-methyl-D-aspartate receptor antagonist, in addition to SSRIs or SNRIs for TRD.

Given the severity of TRD and the uncertain efficacy of available augmentation strategies, new and more-effective drugs for AD augmentation are needed.

A substantial body of research has suggested the possible role of dopamine (DA) dysfunctions in mood disorders: (a) levels of DA are decreased, D2 receptor binding in the striatum is increased, and striatal DA transporter is increased in depression; (b) different DA receptors, mostly D1–D2 heterodimers, and their distribution in different brain region have been suggested to be involved in the etiology of depression; (c) mesolimbic DA neurons could be linked to some nuclear symptoms of depression, such as loss of motivation and motor retardation [13,14].

In addition, some research suggests that the dopamine system plays a critical role in treatment-resistant depression. Patients with treatment-resistant depression typically exhibit severe deficits in interest, motivation, and hedonic capacity, which are thought to be most dependent on the integrity of the dopamine system. Agents that enhance dopamine neurotransmission may therefore be a valuable addition to the therapeutic options available for this condition. For example, monoamine oxidase inhibitors and stimulants, both of which enhance dopamine function, and aripiprazole, a partial dopamine agonist, appear to be effective augmenting agents in treatment-resistant depression [15].

One drug that has been the subject of particular interest in this specific context is pramipexole, a dopamine agonist that has been approved by the FDA for the treatment of Parkinson’s disease and restless legs [16]. The precise mechanisms by which it might exert its antidepressant effects in patients with TRD remain unknown. However, different mechanisms have been proposed, including the following: (a) marked selectivity for D3 receptors, which have a high concentration in mesolimbic areas and are implicated in mental processes related to emotion and mood; (b) dopamine auto-receptor and dopamine transporter interaction; (c) inhibition of NMDA receptors and/or NO-cGMP synthesis; (d) increase in dopamine and serotonin neurotransmission; (e) changes in metabolism in certain brain regions; (f) neuroprotective, antioxidant, and anti-inflammatory activity; and (g) inhibition of aversive information processing [13,17,18,19,20].

A meta-analysis of 18 randomized controlled trials showed that pramipexole improves depressive symptoms in patients with Parkinson’s disease [21], and another meta-analysis of 5 randomized controlled trials and 8 observational studies showed its effectiveness and safety for patients with treatment-resistant and non-resistant depression both in monotherapy and as an augmentation of traditional antidepressants [22].

These preliminary results were confirmed in a later observational study involving only patients with unipolar and bipolar TRD, of which 74.1% achieved response and 66.4% remission after 24 weeks of pramipexole AD augmentation [23].

Overall, the available preliminary data suggest that pramipexole may be useful as an AD augmentation in TRD, but further confirmation is needed. One way to obtain further evidence is to compare the usefulness of pramipexole with that of other drugs that have been shown to be effective as augmentations in the treatment of TRD.

The present observational retrospective study aims to achieve this goal by comparing the 12- and 24-week effectiveness and safety of pramipexole and the long-term (12 and 24 months) sustained response and safety of pramipexole and aripiprazole AD augmentation in the treatment of unipolar TRD. We chose aripiprazole as the comparator because it is considered to be the most effective within the antipsychotic class.

## 2. Methods

### 2.1. Subjects

Data were extracted from a clinical database of consecutive patients seen from September 2006 to August 2021 at the Institute of Psychopathology in Rome, Italy, a private out-patients clinic specializing in mood and anxiety disorders. Inclusion criteria were as follows: (1) age ≥ 18 years; (2) meeting DSM-5 diagnostic criteria for major depressive disorder (MDD), single or recurrent [24]; (3) meeting DSM-5 criteria for a current MDE; (4) a 21-item Hamilton Depression Rating Scale (HDRS_21_) [25] total score ≥ 14; (5) failure of at least 2 trials with 2 antidepressants of separate classes; (6) augmentation of ongoing antidepressant treatment with pramipexole or aripiprazole.

Patients who, at least, improved at 24 weeks were followed for an additional 2 years.

All patients gave their written consent for the use, in anonymous and aggregate form, of their clinical records. The content of the consent form, as well as the procedures of the study, were approved by the local ethical committee (Prot. N. 1521/CE Lazio 1, Roma 30 July 2019) and are in accordance with the Helsinki declaration of 1975, revised in 2013.

### 2.2. Assessments

All subjects were evaluated, treated, and followed up by the first author (AT), an experienced psychiatrist specializing in anxiety and mood disorders. At intake, the mood and anxiety disorders sections of the SCID-I [26,27] were administered to support the clinical diagnosis, and a semi-structured interview was used to systematically collect the participants’ demographic and clinical retrospective information. Whenever possible, patients’ information was supported by secondary clinical data, obtained from other informants or available medical records. During the follow-up visit, which typically occurred each month up to 24 weeks, and subsequently every 2–4 months, information on the course of the illness and treatment was systematically collected from patient and caregivers via a semi-structured interview. At intake and at each follow-up visit, depressive symptoms were assessed using the HDRS_21_, suicidality using item 3 of the HDRS_21_ (score ≤ 1 absent, score ≥ 2 present), (hypo)mania symptoms using the Young Mania Rating Scale (YMRS) [28], clinical status with the Clinical Global Impression of Severity (CGI-s) and of Improvement (CGI-i) scales [29], and the overall level of functioning using the Global Assessment of Functioning (GAF) [30]. The diagnosis of mixed depression was made using the validated Koukopoulos’ criteria [31], consisting of the presence of three or more of the following symptoms during an MDD: (1) psychic agitation or inner tension; (2) racing or crowed thoughts; (3) irritability or unprovoked feelings of rage; (4) absence of retardation; (5) talkativeness; (6) dramatic description of suffering or frequent spell of weeping; (7) mood lability and marked emotional reactivity; and (8) early insomnia.

Treatment adherence was defined as a ≥75% adherence to the prescribed medication regimen according to the patients’ and relatives’ report. 

At each visit, the onset of psychotic symptoms (hallucinations and/or delusions), lethargy, gambling, hypersexuality, and compulsive shopping were explicitly valued in patients on pramipexole augmentation. Other side effects were assessed through clinical observation and patients’ and relatives’ reports. In the medical records, we noted all the reported side effects up to 24 weeks, that required a dose reduction or the discontinuation of the augmentation during the follow-up.

These assessments are made during routine clinical practice at the Institute and not for research purposes. The rating scales were administered by psychiatrists experienced in mood disorders and not involved in the treatment.

### 2.3. Treatments

Pramipexole was added to the current AD regimen starting with 0.18 mg/day in the first week and increasing of 0.18 mg/day every week, with an initial goal of 0.54 mg/day. If the remission was not achieved, the dose was further increased by 0.18 mg/day every 4 days up to 2.1 mg/day, depending on clinical response and the occurrence of significant side effects. Aripiprazole was added to the current AD regimen, starting with 2.5 mg/day and increasing up to 10 mg/day if necessary, depending on clinical response and occurrence of significant side effects. A mood stabilizer (lithium carbonate, valproate, carbamazepine, lamotrigine) was added to pramipexole or aripiprazole augmentation in patients with mixed depression or mood instability. Olanzapine was added to pramipexole augmentation in patients with mood congruent psychotic features when necessary.

All treatments were given as part of routine clinical care and no treatment decision was made with research in mind. Patients and caregivers were informed about the off-label use of pramipexole and its more-dangerous side effects, i.e., psychotic symptoms, lethargy, gambling, hypersexuality, and compulsive shopping.

### 2.4. Outcomes

At 12 and 24 weeks, the primary outcome was remission, defined as a HDRS_21_ total score < 7 after 12 and 24 weeks of treatment and maintained for at least further 4 weeks; secondary outcomes were response, defined as a ≥ 50% reduction of baseline HDRS_21_ total score after 12 and 24 weeks of treatment and maintained for at least further 4 weeks, improvement, defined as a CGI-i score ≤ 2 (“much improved” or “very much improved”) after 12 and 24 weeks of treatment and maintained for at least further 4 weeks, and improvement of functioning, defined as increase in GAF score after 12 and 24 weeks of treatment.

During the follow-up, the primary outcome was the absence of recurrences; secondary outcomes were the percentage of time spent ill during the 12 and 24 months of follow-up, latency of first relapse, defined as number of weeks to first recurrence, and general functioning, defined as median increase in GAF score during the 12 and 24 months of follow-up.

Safety outcomes were acceptability, defined as the number of patients who discontinued the study for any reason, tolerability, defined as the number of patients who discontinued the study due to any side effect, suicidality (defined as HDRS_21_ item 3 ≥ 2) (only at 12 and 24 weeks), suicide attempt, hospitalization, and occurrence of (hypo)mania (ruled out clinically and by the increase in YMRS total score). Patients developing (hypo)mania in the 24 weeks of treatment were considered non responders/remitters.

### 2.5. Statistical Analysis

The sociodemographic and clinical characteristics of patients with pramipexole augmentation (PA) and with aripiprazole augmentation (AA) were summarized as frequencies and percentages for categorical variables, as mean and standard deviation (SD), or median and interquartile range (IQR) for continuous variables, as appropriate.

Patients with PA and with AA were compared using Pearson’s Chi-square test or Fisher’s exact test for categorical variables, t-test for continuous variables with normal distribution, and Mann–Whitney U test for continuous variables with a non-normal distribution. Comparisons of scale scores from baseline to 12 weeks and to 24 weeks, stratified by the type of augmentation, were performed using Friedman’s test in patients with complete data. Bonferroni correction for multiple testing was applied for pairwise comparisons.

To analyze the trend of remission, response, and improvement rates and the trend of GAF scale scores between patients with PA and with AA over time (baseline, 12 weeks, and 24 weeks), repeated measures linear mixed models were used. The main effects investigated were the mean effect of augmentation (PA vs. AA), the time trend of rates/scores regardless of the type of augmentation, and the time trend by group.

To balance the characteristics of patients with PA and with AA, a propensity score was calculated using logistic regression, in which the type of augmentation was regressed on the characteristics that differed significantly between groups at baseline. The predicted probability (i.e., the propensity score) was included as a covariate in the mixed models.

Patients who were, at least, improved at 24 weeks were followed up for 2 years. The proportion of relapsers at 12 months and 24 months was compared between patients with PA and with AA using Fisher’s exact test.

All analyses were carried out by using IBM SPSS, version 25. The significance level was set at 0.05.

## 3. Results 

### 3.1. Study Sample Characteristics and Treatments

Of the 2938 patients with MDD treated from September 2006 to August 2021 at the Institute, 132 meet the inclusion criteria: 104 (78.8%) with recurrent MDD and 28 (21.2%) with a single episode. Table 1 and Table 2 show the demographic and clinical characteristics of the study sample. Briefly, in the overall sample, the median (IQR) duration of the disorder was 192 (60; 318) months, the median (IQR) duration of the index episode was 63 (48; 162) weeks, the median (IQR) number of AD cycles failed was 4 (3; 6), 90 patients (68.2%) also failed to respond to a combination strategy (i.e., to a combination of 2 AD acting on different monoamines), 34 patients (25.8%) failed to an augmentation strategy (i.e., to the addiction of a second agent to the antidepressant regimen), and 1 (in the AA group) to electroconvulsive therapy.

A total of 81 patients (61.4%) received AD pramipexole augmentation and 51 (38.6%) AD aripiprazole augmentation. Compared with patients with AA, those with PA significantly differed in terms of age (older), full-time employment (lower), obsessive compulsive disorder (OCD) (lower) and cerebrovascular disease (higher) comorbidity, age at first episode (higher), duration of the index episode (lower), mixed depression (lower), augmentation strategies previously failed (higher), and HDRS_21_ total scores (higher) and YMRS total scores (lower) at study entry. These variables were used to calculate the propensity score. The median max dose of pramipexole in patients with PA was 1.05 mg/day (IQR 0.72; 1.08), and that of aripiprazole in patients with AA was 3 mg/day (IQR 2.5; 5). As shown in Table 3, the two groups did not differ in terms of antidepressant treatment (classes, doses, and combination strategies) and additional mood stabilizers (classes and doses) employed for the index episode.

### 3.2. Twelve and 24-Week Outcomes

A total of 20 patients (15.2%) dropped out, 8 before 12 weeks (7 with PA, 1 with AA) and 12 before 24 weeks (5 with PA, 7 with AA), without significant differences between the two groups (Exact Fisher’s test *p* = 0.117 and *p* = 0.892, respectively). These 20 patients were considered as non-responders/remitters and did not differ in terms of their demographic or clinical characteristics compared to patients followed-up to 24 weeks.

Treatment adherence ≥ 75% was found in 67 (82.7%) patients with PA and 47 (92.2%) with AA at 12 weeks (*p* = 0.124) and in 57 (70.4%) patients with PA and 41 (80.4%) patients with AA at 24 weeks (*p* = 0.256).

HDRS_21_ median total score significantly decreased from baseline to 12 weeks and to 24 weeks in patients with PA (median [IQR]: 18 [16; 21] vs. 6 [2; 12] vs. 2 [0; 5]; *p* < 0.001) and in patients with AA (median [IQR]: 16.5 [15; 19] vs. 14 [7; 16] vs. 9 [4; 16]; *p* < 0.001).

CGI-s score significantly decreased from baseline to 12 weeks and to 24 weeks in patients with PA (median [IQR]: 4 [4; 5] vs. 2 [2; 3] vs. 2 [1; 2]; *p* < 0.001) and in patients with AA (median [IQR]: 5 [4; 5] vs. 4 [3; 5] vs. 3 [2; 4]; *p* < 0.001).

Figure 1 shows the predicted rates of remission, response, and improvement and the predicted mean GAF score over time and according to the type of augmentation, adjusted for the propensity score, resulting from the repeated measures generalized linear mixed models. In the Appendix A, we report the results of the repeated measures generalized linear mixed models (ESM Appendix A), the predicted rates of remission, response, and improvement, and the predicted mean of GAF score according to the type of augmentation and time, adjusted for the propensity score (ESM Appendix A).

Overall, patients with PA were more likely to remit (OR = 4.24; 95%CI: 1.54–11.67) and respond (OR = 3.77; 95%CI: 1.48–9.60) than patients with AA, while no difference in improvement rate was observed between the two groups (Figure 1).

No significant effects were observed over time and during the interaction term in any of the three outcomes (ESM Appendix A.

In the overall sample, GAF total score significantly increased from baseline to 12 weeks by 12.2 points (*p* < 0.001) and to 24 week by 14.9 points (*p* < 0.001). In patients with PA, the GAF score was, on average, 10.5 points significantly higher than that of patients with AA (*p* = 0.003). Moreover, patients with PA showed a more-rapid increase in GAF scores from baseline to 12 weeks (b = 0.10; *p* = 0.014) and to 24 weeks (b = 0.13; *p* = 0.004) than patients with AA (ESM Appendix A).

### 3.3. Twelve and 24-Week Safety

In the PA group, eight patients dropped out due to the side effects, six before 12 weeks (three due to transient visual hallucinations, one due to edema of the ankles, drowsiness, and increased anxiety, respectively), and two before 24 weeks (due to drowsiness); in the AA group, four patients dropped out due to the side effects, one before 12 weeks (due to increased irritability) and three before 24 weeks (one due to increased anxiety, buccal dyskinesia, and restlessness/tremor, respectively). No differences in the proportion of drop-outs due to side effects between the two groups were observed (Exact Fisher’s test, 12 weeks: *p* = 0.282; 24 weeks: *p* = 0.430).

Other reported side effects in the PA group during the 12 weeks were nine spontaneously resolved (drowsiness and nausea in two cases, respectively; fainting/falling, postural instability, agitation, itching, and dizziness in one case, respectively) and two resolved through pramipexole dose reduction followed by escalation (drowsiness and tremor in one case, respectively). From 12 to 24 weeks, other side effects were four spontaneously resolved (xerostomia in two cases; vomit and walking difficulty in one case, respectively) and one resolved through pramipexole dose reduction (fainting/falling).

In the AA group, five side effects were reported during the 12 weeks (restless in two cases; nausea, somnolence, and dizziness in one case, respectively) and ten from 12 to 24 weeks (weight increase in five cases; xerostomia and tremors in two cases, respectively; urinary difficulties in one case). These side effects required no treatment modification.

The median score of the HDRS_21_ suicidality item significantly decreased from the baseline to 12 weeks and to 24 weeks in patients with PA (baseline: 0 (IQR 0; 1) vs. 12 weeks: 0 (IQR 0; 0) vs. 24 weeks: 0 (IQR 0; 0); *p* < 0.001). In patients with AA, the overall significant decrease (*p* = 0.006) was not confirmed in the pairwise comparisons (baseline: 0 (IQR 0; 1) vs. 12 weeks: 0 (IQR 0; 0) vs. 24 weeks: 0 (IQR 0; 0)).

YMRS median total score did not change significantly from baseline to 12 weeks and to 24 weeks in patients with PA (baseline: 0 (IQR 0; 0) vs. 12 weeks: 0 (IQR 0; 0) vs. 24 weeks: 0 (IQR 0; 0) *p* = 0.220); while in patients with AA, an overall significant decrease was observed (*p* = 0.002), but not in the pairwise comparisons (baseline: 0 (IQR 0; 2) vs. 12-week: 0 (IQR 0; 2) vs. 24 weeks: 0 (IQR 0; 0)).

Before 24 weeks, one patient with PA needed hospitalization due to the severity of the depression. No patients committed attempted suicide or developed (hypo)mania.

### 3.4. Long-Term Outcomes

A total of 60 patients (43 with PA and 17 with AA), at least improved at 24 weeks, were followed for an additional 12 months and 39 (30 with PA and 9 with AA) for an additional 24 months. The drop-out rate did not differ significantly between the PA and AA groups before 12 months (4.7% and 17.6%, respectively; *p* = 0.132) and 24 months (3.3% and 11.1%, respectively; *p* = 0.413). Treatment adherence ≥ 75% was found in 32 (74.4%) patients with PA and 11 (64.7%) with AA at 12 months and 25 (83.3%) patients with PA and 7 (77.8%) with AA at 24 months, without significant difference between the two groups (*p* = 0.547 and 0.653, respectively).

At 12 months, 34 out of 43 patients with PA (79.1%) and 9 out of 17 patients with AA (52.9%) did not relapse; at 24 months, 24 out of 30 patients with PA (80.0%) and 6 out of 9 patients with AA (67.7%) did not relapse. No difference in the probability of relapse was found between patients with PA and AA at each time point. The polarity of the first relapse was depression in eight patients with PA and six with AA; mixed depression in one patient with PA and one with AA; hypomania in one patient with AA at 12 months; depression in three patients with PA and three patients with AA; mixed depression in two patients with PA; and hypomania in one patient with PA at 24 months.

No difference in the latency of the first relapse was found between the PA group and the AA group at 12 months (median 40 (IQR 10; 60) and 20 (IQR 9; 30) weeks, respectively) and at 24 months (median 88 (IQR 76; 100) and 52 (IQR 45; 88) weeks, respectively).

The number of relapses was 10 in the PA group (9 depression, 1 mixed depression) and 13 in the AA group (12 depression, 1 mixed depression, 1 hypomania) at 12 months; 7 (3 depression, 3 mixed depression, 1 hypomania) in the PA group and 4 (all depression) in the AA group at 24 months.

The percentage of time spent on illness was significantly lower in the PA group than in the AA group at 12 months (median 0% (IQR 0%; 0%) vs. 0% (IQR 0%; 22%), respectively; *p* = 0.022) but not at 24 months (median 0% (IQR 0%; 0%) vs. 0% (IQR 0%; 14%, respectively; *p* = 0.566).

The median GAF score was significantly higher in the PA group than in the AA group at 12 months (77 (IQR 71; 91) and 65 (IQR 58; 71), respectively; *p* < 0.001) but not at 24 months (81 (IQR 74; 84) and 71 (71;(IQR 71; 71), respectively; *p* = 0.072).

### 3.5. Long-Term Safety

Before 12 months, one patient with PA dropped-out due to the side effects (increased sexual drive); from 12 to 24 months, no patient dropped out due to the side effects.

Moreover, no patient in both groups needed augmentation’ dose reduction due the side effects, hospitalization, or attempted suicide during follow-up.

As reported in the earlier section, two patients suffered a hypomania episode during the follow-up, one with AA (before 12 months) and one with PA (from 12 to 24 months).

## 4. Discussion

To the best of our knowledge, this is the first study to compare the short- and long-term outcomes of pramipexole and aripiprazole augmentation in patients with unipolar depression not responding to at least two trials with two antidepressants of separate classes.

Overall, our findings indicate that both augmentation strategies improved the level of functioning and were safe, but PA had higher effectiveness. In fact, the PA response and remission rates were almost twice those of AA; the predicted response rates were 64.1% and 32.2% at 12 weeks and 76.2% and 38.0% at 24 weeks, respectively; the predicted remission rates were 49.7% and 18.9% at 12 weeks and 72.7% and 28.1% at 24 weeks, respectively. Furthermore, patients with PA, compared with those with AA, achieved a greater and faster improvement of psychosocial functioning (measured using the GAF score) from baseline to 12 and 24 weeks.

Notably, the response and remission rates for PA in our study are in the range of those reported in previous studies on unipolar patients with TRD (25% to 76% response and 46% to 71% remission) [32], and they are at the lower bottom of the range for AA (32.4% to 43.1% response and 25.4% to 44% remission) [33,34]. The difference in depression resistance—to at least two ADs of separate classes in our study and to at least one AD in the most previous studies—could explain why our response/remission rates to AA are at the lower bottom of the range reported in the literature. Although the median max dose of aripiprazole prescribed as an augmentation in our study (3 mg/day) is at the lower bottom of the dose’s range currently recommended in the international guidelines (2–15 mg/day), it corresponds to the maximum target dose–efficacy curve (2–5 mg/day) shown in a recent meta-analysis, which aimed to find the optimal dose of aripiprazole augmentation [35]. Therefore, we can reasonably discount the possibility that AA response/remission rate in our study depends on the aripiprazole’ dose employed.

PA and AA did not significantly differ on safety outcomes at 12 and 24 weeks and showed a moderate acceptability and tolerability. Side effects were limited in number and severity and resolved spontaneously or through dose reduction. No patients attempted suicide or developed (hypo)mania symptoms, and one patient, in the PA group, needed hospitalization.

Notably, the drop-out rate due to side effects in our study is quite similar to that reported in previous studies for PA in patients with unipolar TRD (7.4% and 6.4%, respectively) [32] and in studies employing an aripiprazole dosage corresponding to the maximum target dose of 4 mg/day for AA in patients with MDD (2% vs. 4%, respectively) [35].

Our findings highlight the higher effectiveness of PA over AA as a maintenance strategy for TRD, at least up to 12 months. In fact, in the first 12 months of follow-up, patients with PA, compared with patients with AA, spent less time ill and achieved a greater improvement in their psychosocial functioning. The two augmentations showed a moderate and similar safety during the follow-up; no patients needed hospitalization or attempted suicide; one dropped-out due to side effects (PA group); and two dropped out due to the development of a hypomania episode (one in the AA and one in the PA group).

Notably, our results are consistent with the very limited data available on the topic, showing that PA and AA retain their effectiveness without losing the safety during the follow-up [15,36].

The main study limitations are as follows: (1) the flexibility of the add-on schedule, which, in some cases, implied the partial change of concomitant treatment strategies on the clinical judgement of the treating clinician; and (2) the clinical assessment of side effects that could reduce the number of less severe side effects reported and, in the follow-up, noted in the medical records. Yet, despite these limitations, it is our opinion that the findings of the present study provide valuable information to clinicians with which to improve the treatment strategy for TRD, as well as to researchers because they open up new perspectives for investigating the neurobiological basis of this highly prevalent clinical condition.

It is important to keep in mind that the evidence in this study is preliminary and requires further confirmation. Furthermore, the use of pramipexole as an AD augmentation for TRD is off-label; its prescription should be reserved for carefully selected patients with severe TRD treated in a specialized center, and its use requires caution. In particular, patients and their caregivers must be advised of potentially dangerous side effects, such as psychotic symptoms, lethargy, gambling, hypersexuality, and compulsive shopping, and instructed to report them immediately when they occur. Furthermore, the dose of pramipexole must be gradually escalated, and its abrupt discontinuation should be avoided to prevent dopamine withdrawal syndrome.

## 5. Conclusions

In conclusion, our study shows that PA is a promising strategy for the short- and long-term treatment of unipolar TRD. Though similar in terms of safety, it is more effective than AA, which currently is the augmentation strategy for TRD with the best evidence for efficacy.

## Figures and Tables

**Figure 1 biomedicines-12-02064-f001:**
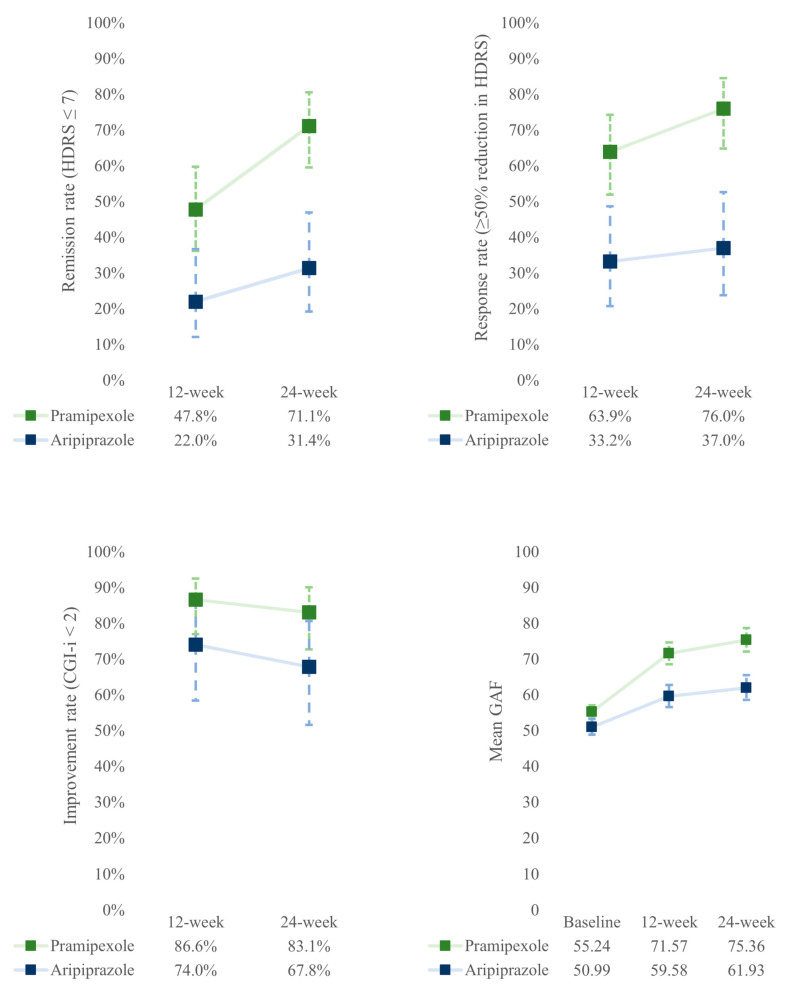
Trend of predicted rates of remission, response, and improvement and predicted mean GAF score according to type of augmentation, adjusting by propensity score. Results obtained from the repeated measures generalized linear mixed models.

**Table 1 biomedicines-12-02064-t001:** Sociodemographic and clinical features of the study sample at baseline.

	Total (n = 132)		Type of Augmentation				Test	*p*-Value
			Pramipexole (n = 81)		Aripiprazole (n = 51)			
	n	%	n	%	n	%		
Female (n, %)	77	58.3%	49	60.5%	28	54.9%	0.40 §	0.526
Age at baseline (yrs), mean (DS)	61.2 (15.1)		65.4 (12.5)		54.6 (16.6)		4.26 °	**<0.001**
Employed full time (n, %)	48	36.4%	23	28.4%	25	49.0%	5.75 §	**0.016**
Married (n, %)	82	62.1%	55	67.9%	27	52.9%	2.98 §	0.084
Diagnosis							0.27 §	0.605
*Major depressive disorder*, *recurrent*	104	78.8%	65	80.2%	39	76.5%		
*Major depressive disorder*, *single episode*	28	21.2%	16	19.8%	12	23.5%		
Any life-time Axis I comorbidity (n, %)	85	64.4%	48	59.3%	37	72.5%	2.41 §	0.121
OCD (n, %)	51	38.6%	24	29.6%	27	52.9%	7.17 §	**0.007**
PD (n, %)	35	26.5%	21	25.9%	14	27.5%	0.04 §	0.847
SP (n, %)	21	15.9%	11	13.6%	10	19.6%	0.85 §	0.357
GAD (n, %)	25	18.9%	16	19.8%	9	17.6%	0.09 §	0.764
ED (n, %)	9	6.8%	3	3.7%	6	11.8%	3.20 §	0.074
Cerebrovascular disease (n, %)	29	22.0%	23	28.4%	6	11.8%	5.05 §	**0.025**
Alcohol abuse (n, %)	11	8.3%	7	8.6%	4	7.8%	*	1.000
Substance abuse (n, %)	4	3.0%	3	3.7%	1	2.0%	*	1.000
Suicide attempts (n, %)	18	13.6%	11	13.6%	7	13.7%	0.01 §	0.981
Hospitalization (n, %)	30	22.7%	19	23.5%	11	21.6%	0.06 §	0.801
Age at first episode (yrs), mean (DS)	42.9 (18.8)		47.1 (18.6)		36.3 (17.3)		3.34 °	**0.001**
Duration of disease (months), median (IQR)	192 (60; 318)		168 (60; 348)		204 (60; 288)		2032.5 #	0.877

Abbreviations: OCD = obsessive compulsive disorder; PD = panic disorder; SP = social phobia; GAD = general anxiety disorder; ED = eating disorder. Data shown in bold are statistically significant. § Pearson’s Chi-square test; * Fisher’s exact test ° *t*-test; # Mann–Whitney U test.

**Table 2 biomedicines-12-02064-t002:** Description of the index episode and the AD treatments failed by groups.

	Total (n = 132)		Type of Augmentation				Test	*p*-Value
			Pramipexole (n = 81)		Aripiprazole (n = 51)			
	n	%	n	%	n	%		
Duration of episode (weeks), median (IQR)	63 (47; 162)		54 (45; 108)		108 (54; 162)		2512.0 #	**0.036**
Mixed depression	34	25.8%	13	16.0%	21	41.2%	10.33 §	**0.001**
Delusional depression							*	0.280
*Yes*, *congruous delusions*	5	3.8%	3	3.7%	2	3.9%		
*Yes*, *incongruous delusions*	2	1.5%	0	0.0%	2	3.9%		
AD treatments failed								
N. of AD cycles failed, median (IQR)	4 (3; 6)		4 (3; 5)		4 (3; 6)		2120.0 #	0.795
SSRI (n, %)	102	77.3%	61	75.3%	41	80.4%	0.46 §	0.497
SNRI (n, %)	96	72.7%	60	74.1%	36	70.6%	0.19 §	0.661
TCA (n, %)	86	65.2%	51	63.0%	35	68.6%	0.44 §	0.506
MAOI (n, %)	5	3.8%	1	1.2%	4	7.8%	*	0.073
Other (n, %)	67	50.8%	45	55.6%	22	43.1%	1.93 §	0.165
ECT (n, %)	1	.8%	0	0.0%	1	2.0%	*	0.386
Combination (n, %)	90	68.2%	60	74.1%	30	58.8%	3.36 §	0.067
Augmentation (n, %)	34	25.8%	30	37.0%	4	7.7%	*	**<0.001**
Baseline scores, median (IQR)								
*HDRS*	18 (16; 20)		18 (16; 21)		17 (16; 19)		1603.5 #	**0.044**
*HDRS-3*	0 (0; 1)		0 (0; 1)		0 (0; 1)		2034.5 #	0.861
*YMRS*	0 (0; 0)		0 (0; 0)		0 (0; 2)		2584.5 #	**0.001**
*CGI-s*	5 (4; 5)		4 (4; 5)		5 (4; 5)		2353.5 #	0.150
*GAF*	55 (50; 55)		55 (50; 60)		50 (50; 55)		1954.5 #	0.587

Abbreviations: SSRI = selective serotonin reuptake inhibitor; SNRI = serotonin noradrenaline reuptake inhibitor; TCA = tricyclic; MAOI = MonoAmine Oxidase Inhibitor; ECT= electroconvulsive therapy; AD = antidepressant. Data shown in bold are statistically significant. § Pearson’s Chi-square test; * Fisher’s exact test; # Mann–Whitney U test.

**Table 3 biomedicines-12-02064-t003:** Treatment of the index episode.

	Total (n = 132)		Type of Augmentation				Test	*p*-Value
			Pramipexole (n = 81)		Aripiprazole (n = 51)			
	n	%	n	%	n	%		
SSRI (n, %)	43	32.6%	23	28.4%	20	39.2%	1.67 §	0.196
SSRI dose, median (IQR)	40 (25; 60)		30 (20; 60)		40 (40; 60)		290.5 #	0.134
SNRI (n, %)	39	29.5%	26	32.1%	13	25.5%	0.66 §	0.418
SNRI dose, median (IQR)	150 (150; 225)		150 (150; 225)		150 (120; 225)		145.0 #	0.489
Others (n, %)							5.66 §	0.059
*Mirtazapina*	20	15.2%	17	21.0%	3	5.9%		
*Amisulpiride*	15	11.4%	8	9.9%	7	13.7%		
TCA (n, %)	70	53.0%	44	54.3%	26	51.0%	0.14 §	0.708
TCA dose, median (IQR)	125 (100; 175)		125 (100; 187)		125 (100; 162)		517.0 #	0.500
2 AD (n, %)	35	26.5%	24	29.6%	11	21.6%	1.04 §	0.307
Mood stabilizer (n, %)							*	0.168
*Lithium*	9	6.8%	6	7.4%	3	5.9%		
*AV/CBZ*	15	11.4%	5	6.2%	10	19.6%		
*LMT*	6	4.5%	4	4.9%	2	3.9%		
*OLZ*	1	.8%	1	1.2%	0	0.0%		
Mood stabilizer dose, median (IQR)								
*Lithium*	0.50 (0.40; 0.59)		0.55 (040; 0.59)		0.45 (0.30; 0.56)		5.5 #	0.381
*VA/CBZ*	300 (300; 450)		300 (250; 300)		300 (300; 600)		34.5 #	0.254
*Li*	200 (200; 200)		200 (175; 225)		200 (200; 200)		4.0 #	1.000
*OLZ*	10 (-)		10 (-)					

Abbreviations: SSRI = selective serotonin reuptake inhibitor; SNRI = serotonin noradrenaline reuptake inhibitor; TCA = tricyclic; AD = antidepressant; VA/CBZ= Valproic acid/ carbamazepine; Li= lithium carbonate; OLZ= olanzapine:. Data shown in bold are statistically significant. § Pearson’s Chi-square test; * Fisher’s exact test; # Mann–Whitney U test.

## Data Availability

Data are available upon reasonable request to the corresponding author.

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
