# Peer review of "Comparative Short- and Long-Term Effectiveness and Safety of Pramipexole and Aripiprazole Augmentation in Treatment-Resistant Unipolar Depression: An Observational Study"

_biomedicines, 2024, doi:10.3390/biomedicines12092064_

Round 1
Reviewer 1 Report
Comments and Suggestions for Authors
The research design of this manuscript is rigorous. The data sources are reliable and the analysis is accurate. The following three questions require further clarification:
1. What questions in current research or practice does this study respond to as the first comparative study? Please specify the significance of the study in more details.
2. The references in this study are so outdated that it is difficult to judge its novelty. Please add more discussion of research from the last 3 years.
3. Why is there so little attention paid to this issue in all the previous studies? Is there an urgent practical need to answer this question?
Author Response
The research design of this manuscript is rigorous. The data sources are reliable and the analysis is accurate. The following three questions require further clarification:
- What questions in current research or practice does this study respond to as the first comparative study? Please specify the significance of the study in more details.
Authors reply: Thank you very much for your comment that allowed us to improve the manuscript. As requested, the objective of the study has been better defined in the Introduction section (lines 82-87)
Overall, the available preliminary data indicate that pramipexole may be useful as AD-augmentation for TRD but further confirmation is needed. One way to obtain further evidence is to compare the usefulness of pramipexole with that of other drugs proven effective as augmentation for the treatment of TRD.
The present study observational, retrospective study aims to achieve this goal by comparing the 12- and 24-week effectiveness and safety…
and the potential utility of the study results for clinicians and researchers has been more clearly articulated in the Discussion section (lines 384-387)
Yet, despite these limitations, it is our opinion that the findings of the present study provide valuable information to clinicians to improve the treatment strategy for TRD, and to researchers because they open up new perspectives for investigating the neurobiological basis of this highly prevalent clinical condition.
- The references in this study are so outdated that it is difficult to judge its novelty. Please add more discussion of research from the last 3 years.
Authors reply: We acknowledged the reviewer's recommendation to remove the outdated references in the Discussion section (line 396) and References section. We now refer to a recent literature review that provides a summary of the evidence on response and remission rates to pramipexole AD augmentation in unipolar TRD (see Reference 15, figures 1 and 2).
- Why is there so little attention paid to this issue in all the previous studies? Is there an urgent practical need to answer this question?
Authors reply: TRD is a prevalent clinical condition with significant personal and social impacts. The multiple treatment strategies available are ineffective in a considerable proportion of cases, with about 50% of treated patients failing to achieve remission. Consequently, searching alternative treatment strategies for these patients is an urgent practical need.
In the last two decades, clinicians have published observational studies showing that pramipexole, when used in combination with conventional antidepressants, may prove efficacious in this patient population.
However, this topic has not been given sufficient attention by clinicians, who could have provided additional data from the real word, because they are discouraged by the off-label use of a drug that can cause serious side effects in patients with Parkinson’s disease (mostly, hallucinations/delusions, lethargy and impulse control).
In addition, researchers have not designed placebo-controlled research, due to the difficulty in obtaining funding from companies (licence expired) and/or because the limited number of published controlled studies yielded less encouraging outcomes than observational studies, due to methodological shortcomings (for example, drug titration, maximum dosage, patient selection, treatment duration) (see Reference 22, pg 121).
Reviewer 2 Report
Comments and Suggestions for Authors
This study compares the short- and long-term effectiveness and safety of pramipexole augmentation (PA) and aripiprazole augmentation (AA) for unipolar treatment-resistant depression (TRD). The results indicated that treatment effect with PA was significantly superior than those with AA after 12 and 24 weeks. The current research topic is meaningful and the method is scientifically sound. Some minor problems require further consideration by the author.
First, the mechanism of PA in treating unipolar treatment-resistant depression needs further elucidation.
Second, some formatting issues need to be corrected, for example, the underline in Table 2 and 3 were missing.
Author Response
This study compares the short- and long-term effectiveness and safety of pramipexole augmentation (PA) and aripiprazole augmentation (AA) for unipolar treatment-resistant depression (TRD). The results indicated that treatment effect with PA was significantly superior than those with AA after 12 and 24 weeks. The current research topic is meaningful and the method is scientifically sound. Some minor problems require further consideration by the author.
First, the mechanism of PA in treating unipolar treatment-resistant depression needs further elucidation.
Authors reply: Thank you very much for your comment that allowed us to improve the manuscript. In the Introduction section (lines 53-61) we further explain the rationale for the use of dopamine agonist in patients with TRD
In addition, some research indicates that the dopamine system plays a crucial role in treatment-resistant depression. Patients with treatment-resistant depression typically exhibit profound deficits in interest, motivation and hedonic capacity, which are thought to be most dependent on the integrity of the dopamine system. Agents that enhance dopamine neurotransmission may therefore be a valuable addition to the therapeutic options available for this condition. For example, monoamine oxidase inhibitors and stimulants, both of which enhance dopamine function, and aripiprazole, a partial dopamine agonist, appear to be effective augmenting agents in treatment-resistant depression (15)
elucidate the potential mechanisms by which pramipexole might exert its antidepressant effects in patients with TRD (lines 62-72)
One drug that has been the subject of particular interest in this specific context is pramipexole, a dopamine agonist that has been approved by the FDA for the treatment of Parkinson's disease and restless legs (16) . The precise mechanisms by which it might exert its antidepressant effects in patients with TRD remain unknown. However, different mechanisms have been proposed, including: a) marked selectivity for D3 receptors, which have a high concentration in mesolimbic areas and are implicated in mental processes related to emotion and mood; b) dopamine auto-receptor and dopamine transporter interaction; c) inhibition of NMDA receptors and/or NO-cGMP synthesis, d) increase in dopamine and serotonin neurotransmission, e) changes in metabolism in certain brain regions, f) neuroprotective, antioxidant and anti-inflammatory activity, and g) inhibition of aversive information processing [13; 17-20]
and added the new reference to the Reference list
- Martens AAG, Kaltenboeck A, Halahakoon et al. An experimental medicine investigation of the effects of subacute pramipexole treatment on emotional information processing in healthy volunteers. Pharmaceuticals (Basels) 2021 Aug 14;14(8):800. doi: 10.3390/ph14080800.
Second, some formatting issues need to be corrected, for example, the underline in Table 2 and 3 were missing.
Authors reply: we thank the reviewer for noticing these inaccuracies. and reformatted tables 2 and 3.
Reviewer 3 Report
Comments and Suggestions for Authors
A very good manuscript on the Pramipexole and Aripiprazole relevance in
Resistant Unipolar Depression
Paper is modern and can be of interest for the readers
Introduction sets the scene very well, is up to date and i have no comments here
Same methodology is very nice described
REsults are complex, and, as stated, could have some relevance in the field and for the readers of the journal
Discussion is very well constructed
I would only keep the first part of the conclusions, stating the relevant findings. the second paragraph of the conclusions section seems to belong to the discussion section, rather then the conclusions section
Paper can be accepted
Comments on the Quality of English Language
english is good
Author Response
A very good manuscript on the Pramipexole and Aripiprazole relevance in
Resistant Unipolar Depression
Paper is modern and can be of interest for the readers
Introduction sets the scene very well, is up to date and i have no comments here
Same methodology is very nice described
Results are complex, and, as stated, could have some relevance in the field and for the readers of the journal
Discussion is very well constructed
I would only keep the first part of the conclusions, stating the relevant findings. the second paragraph of the conclusions section seems to belong to the discussion section, rather then the conclusions section
Authors reply: we thank the reviewer for noticing this point. As suggested, the following paragraph has been moved from the Conclusion to the Discussion section (lines 389-397)
It is important to keep in mind that the evidence from this study is preliminary and warrants further confirmation. Furthermore, the use of pramipexole as AD-augmentation for TRD is off-label, its prescription should be reserved only to carefully selected patients with severe TRD treated in specialized center and its use need several cautions. Especially, patients and their caregivers must be advised of potentially dangerous side effects as psychotic symptoms, lethargy, gambling, hypersexuality and compulsive shopping and instructed to report it immediately when they occur. Furthermore, the dose of pramipexole must be gradually escalated and its abrupt discontinuation should be avoided to prevent dopamine withdrawal syndrome.
Paper can be accepted